# An Integrated Computational and Experimental Approach to Formulate Tamanu Oil Bigels as Anti-Scarring Agent

**DOI:** 10.3390/ph17010102

**Published:** 2024-01-11

**Authors:** Megha Krishnappa, Sindhu Abraham, Sharon Caroline Furtado, Shwetha Krishnamurthy, Aynul Rifaya, Yahya I. Asiri, Kumarappan Chidambaram, Parasuraman Pavadai

**Affiliations:** 1Department of Pharmaceutics, Faculty of Pharmacy, M.S. Ramaiah University of Applied Sciences, Gnanagangothri Campus, Bengaluru 56054, Karnataka, India; meghakrishnappa23@gmail.com (M.K.); sharongonsalves2006@gmail.com (S.C.F.); shwek2014@gmail.com (S.K.); 2Department of Chemical Engineering, Erode Sengunther Engineering College, Erode 638057, Tamil Nadu, India; aynulbioinfo@gmail.com; 3Department of Pharmacology, College of Pharmacy, King Khalid University, Abha 61421, Asir Province, Saudi Arabia; yialmuawad@kku.edu.sa; 4Department of Pharmaceutical Chemistry, Faculty of Pharmacy, M.S. Ramaiah University of Applied Sciences, Gnanagangothri Campus, Bengaluru 560054, Karnataka, India

**Keywords:** bigel, *Calophyllum inophyllum*, calanolide A, piscean collagen, tamanu oil, molecular docking, ADMET

## Abstract

Tamanu oil has traditionally been used to treat various skin problems. The oil has wound-healing and skin-regenerating capabilities and encourages the growth of new skin cells, all of which are helpful for fading scars and hyperpigmentation, as well as promoting an all-around glow. The strong nutty odor and high viscosity are the major disadvantages associated with its application. The aim of this study was to create bigels using tamanu oil for its anti-scarring properties and predict the possible mechanism of action through the help of molecular docking studies. In silico studies were performed to analyze the binding affinity of the protein with the drug, and the anti-scarring activity was established using a full-thickness excision wound model. In silico studies revealed that the components inophyllum C, 4-norlanosta-17(20),24-diene-11,16-diol-21-oic acid, 3-oxo-16,21-lactone, calanolide A, and calophyllolide had docking scores of −11.3 kcal/mol, −11.1 kcal/mol, −9.8 kcal/mol, and −8.6 kcal/mol, respectively, with the cytokine TGF-β1 receptor. Bigels were prepared with tamanu oil ranging from 5 to 20% along with micronized xanthan gum and evaluated for their pH, viscosity, and spreadability. An acute dermal irritation study in rabbits showed no irritation, erythema, eschar, or edema. In vivo excisional wound-healing studies performed on Wistar rats and subsequent histopathological studies showed that bigels had better healing properties when compared to the commercial formulation (Murivenna^TM^ oil). This study substantiates the wound-healing and scar reduction potential of tamanu oil bigels.

## 1. Introduction

Damaged skin is repaired through wound healing, a complex biological process that includes hemostasis, inflammation, proliferation, and remodeling. The final three steps are essential in determining if healing is taking place naturally or if it is causing excessive extracellular matrix (ECM) protein production and fibrosis, which would indicate an aberrant healing [1]. The restoration of skin barrier function after wounding or damage plays a major role in preventing further damage to the skin. Extended wound healing may even hinder normal wound healing, resulting in scarring [1]. Scars are the normal and inevitable outcome of mammalian tissue repair. Tissue regeneration, as well as the formation of new tissue similar to the original undamaged skin, is the ideal endpoint [2]. Scar formation may be due to the overproduction of connective tissue (collagen) and differences during the wound-healing process. Millions of people experience scarring annually as a result of burns, trauma, or skin injury following surgery, which has an impact on mental health [3]. Keloid and hypertrophic scars are a prevalent issue as wound healing progresses. Clinically, they are distinguished by an excessive buildup of collagen due to damage to the dermis and subcutaneous regions. Growth factors and cytokines include transforming growth factor β (TGF-β), epidermal growth factor (EGF), fibroblast growth factor (FGF), and platelet-derived growth factor (PDGF) that control this process [4].

The cytokine transforming growth factor β (TGF-β), which is released by many cell types involved in wound healing, is essential for the healing process. TGF-β comprises three isomers, TGF-β1, TGF-β2, and TGF-β3, which together influence fibroblast proliferation, angiogenesis, and ECM synthesis by promoting the infiltration of inflammatory cells. In addition, TGF-β prevents re-epithelialization [5,6]. The effects of various TGF-β isoforms on wound healing may vary depending on the environment. TGF-β3 has also been reported to facilitate reduced scarring, but TGF-β1 may also mediate fibrosis in adult wounds. In order to treat both acute and chronic wounds as well as fibrosing illnesses, TGF-β3 may provide a scar-reducing therapy [7].

Since ancient times, tamanu oil, which is extracted from the nuts of the tamanu tree (*Calophyllum inophyllum* Linn.), has been used for a variety of skin-care purposes. The plant is found primarily in Australia, East Africa, India, Malaysia, and the Pacific Ocean. The oil is dark green in color, with a nutty odor and disagreeable taste. Owing to the non-fatty constituents present in the oil, it is not edible. Tamanu oil has limited scientific evidence, although it is widely used to treat scars, burns, diabetic wounds, and abrasions [8]. The oil is reported to contain constituents such as calophyllolide, inophyllum C, inophyllum D, inophyllum E, inophyllum P, tamanolide D, tamanolide P, calanolide A, calanolide B, and calanolide D. Calophyllolide, the major constituent, is responsible for anti-coagulant, anti-inflammatory, anti-aging, wound-healing, and antioxidant and anti-microbial properties [9,10,11,12,13]. Tamanu oil is also effective in scar removal, reduction of stretch marks, and treatment of psoriasis, eczema, skin allergies, sunburn, and acne. Extracts of the stem bark and seeds of *Calophyllum inophyllum* have also shown promising anti-arthritic activity in a Freund’s-complete-adjuvant-induced arthritis model in rats [14]. Calophyllolide, calophyllic acid, and inophyllum, as well as polyphenols with antioxidant properties, are components of *Calophyllum inophyllum* that are responsible for the ability of the plant to promote wound healing. However, the medicinal benefits of tamanu oil have only been reported and quantified in a small number of scientific investigations [15,16,17,18,19]. The oil has wound-healing and skin-regenerating capabilities and encourages the growth of new skin cells, all of which are helpful for fading scars and hyperpigmentation, as well as promoting an all-around glow [20].

Oleogels and hydrogels, two distinctive solid-like formulations with useful qualities for cosmetic and medicinal applications, are combined to create a bigel. Bigel exhibits superior qualities to individual gels and is capable of delivering both hydrophilic and lipophilic active substances [2,21]. They are different when compared to creams and ointments as they are devoid of surfactants, enhance hydration of the stratum corneum, resulting in cooling and moisturizing effects, enhance permeability of drugs, can be prepared easily, and are water-washable [22,23].

The purpose of this study was to investigate the potential of tamanu oil as an active ingredient for the development of bigels with significant scar-reducing properties. An in silico study elucidating its interaction with 5E8W (TGF-β receptor type-1), a membrane-bound TGF beta receptor protein, was performed to demonstrate its clinical potential. The scar-free wound-healing potential of tamanu oil was compared with that of Ayurvedic murivenna oil, which is recommended by Ayurvedic physicians for its wound-healing and anti-inflammatory properties [24,25,26].

## 2. Results and Discussion

### 2.1. GC-MS Characterization of Tamanu Oil

The chemical composition of tamanu oil was determined by GC-MS and compared with previously reported data. The analysis revealed the presence of the main components calanolide A, calophyllolide (inophyllum derivative), inophyllum C, 4-norlanosta-17(20),24-diene-11,16-diol-21-oic acid, 3-oxo-16,21-lactone (steroid lactone), along with fatty acids palmitic acid, oleic acid, linoleic acid, stearic acid, and hyenic acid (Table 1 and Appendix A). The retention times of the components were identified by comparison with reported literature [18]. The gas chromatogram and corresponding mass spectra are shown in Appendix A. These constituents have been reported to possess anti-microbial, anti-inflammatory, antioxidant, wound-healing, anti-HIV, and anti-tumor activities [27,28,29,30]. Calophyllolide exhibits wound-healing activity by reducing myeloperoxidase (MPO) activity and regulating inflammatory cytokines (IL-1β, IL-6, and TNF-α). The pro-inflammatory cytokines IL-1β, IL-6, and TNF-α are downregulated, whereas the anti-inflammatory cytokine IL-10 is upregulated by calophyllolide [18].

### 2.2. In Silico Studies—Molecular Modeling

The extracellular matrix (ECM) is modulated by transforming growth factor beta (TGF-β), which increases collagen synthesis and controls the expression of numerous genes encoding the extracellular-matrix-degrading matrix metalloproteinases (MMPs) [31]. Transforming growth factor beta 1 (TGF-β1) is a polypeptide member of the TGF-β superfamily of cytokines that contributes to the progress of scar formation through its stimulatory effects on the manifestation of key ECM components and its inhibitory effects on the expression of MMPs in fibroblasts. Additionally, it stimulates collagen production in fibroblasts [32].

Initially, molecular docking studies were carried out to explore the possible synergetic mechanism of tamanu oil ingredients with the various protein targets that play a major role in scar formation. Binding energy for the selected targets is shown in Appendix A. A comprehensive investigation of binding energy and binding interactions drove the eventual choice of the target molecule. The goal of this critical evaluation was to determine the strength and specificity of molecular interactions between the target molecule and its anticipated binding partners. Binding energy was the most important factor in determining the stability of these interactions, ensuring that the selected target (TGF protein) had strong and favorable binding features. Furthermore, a careful investigation of binding interactions revealed information about the kind of bonds produced between the target molecule and its binding partners. This thorough review sought to discover a target molecule with appropriate binding characteristics, focusing on both the affinity and specificity of molecular interactions with TGF protein. Hence, TGF protein was selected for further analysis. To analyze the binding and molecular interactions of the active constituents of tamanu oil with the potential therapeutic target TGF-β type 1 kinase domain (T204D) in complex with staurosporine (PDB ID:5E8W) [33], molecular docking studies were performed using PyRx 0.8 [34,35,36,37]. The predicted binding affinities of all identified constituents, as well as the staurosporine standard, was found to be −5.0 kcal/mol to −11.3 kcal/mol for test compounds and -8.6 kcal/mol for standard, as presented in Table 2.

In accordance with the findings of the molecular docking experiments, the chosen standard staurosporine and the bioactive compounds inophyllum C, 4-norlanosta-17(20),24-diene-11,16-diol-21-oic acid, 3-oxo-16,21-lactone, calanolide A, and calophyllolide exhibited distinct patterns of interaction with the target protein. Staurosporine, with a binding score of −8.6 kcal/mol, exhibited many interactions. These included hydrophobic contacts with amino acids Ile 211, Gly212, and Lys337; hydrogen bond interactions with Ser280, Asp281, and His 283; and π-stacking interactions with Val219, Ala230, Lys232, Leu340, and Ala350. Inophyllum C demonstrated improved binding, including H-bond interactions with His283 and hydrophobic interactions with Val219, Ala230, Lys232, Leu260, Leu340, Ala350, and Asp351. Furthermore, 4-norlanosta-17(20), 24-diene-11,16-diol-21-oic acid, 3-oxo-16,21-lactone had a significant binding score of −11.0 kcal/mol. Salt bridges, hydrophobic contacts, and hydrogen bond interactions were all generated as a result of these interactions. Both calanolide A and calophyllolide exhibited distinct interactions, such as hydrophobic and H-bond interactions, with binding scores of −9.8 kcal/mol and −8.6 kcal/mol when compared to one another. In addition to providing substantial insights into the binding processes and probable pharmacological importance of these chemicals, these findings also pave the way for further study and the possibility of therapeutic uses. Figure 1a–j illustrates the visualization of the interactions that occurred between the chosen bioactive chemicals and the standard with the protein. This visualization was carried out with the help of Discovery Studio Visualizer.

According to Lipinski’s rule of five, molecules with poor permeation will have molecular weight ˃ 500, log *p* ˃ 5, hydrogen bond donors ˃ 5, and hydrogen bond acceptors ˃ 10. Among all the ligands investigated, calanolide A showed a molecular weight of 370.445, log *p* of 4.3801, one hydrogen bond donor, and five hydrogen bond acceptors, suggesting good permeation characteristics (Appendix A).

The absorption, distribution, metabolism, excretion, and toxicity (ADMET) of the constituents of tamanu oil were predicted using in silico methods (Appendix A) [38].

### 2.3. Formulation of Bigels

Eight formulations containing 5–20% tamanu oil were prepared. Formulations BG1–BG4 contained 1% micronized xanthan gum in the hydrogel phase, whereas BG5–BG8 contained 2% micronized xanthan gum in the hydrogel phase. The concentrations of Tween 20 and Geogard^®^ ECT were maintained at 3% and 1%, respectively, for all the formulations. The nutty odor of tamanu oil was masked with vanilla fragrance oil.

### 2.4. Evaluation of Bigel

All formulations were pastel green with a mild nutty odor, non-greasy, shiny texture, and easy washability and were homogenous with good consistency. The fragrances of topical products can significantly affect customer acceptance and satisfaction. As a result, smell is an essential characteristic of cosmetics that consumers consider and appreciate during their selection [39]. The strong odor of the oil was masked by vanilla fragrance.

#### 2.4.1. pH

The regular use of cosmetics can help maintain skin health by controlling the pH of skin. In some skin disorders, topical products that correct skin pH should be part of the treatment plan. Therefore, it is crucial to carefully consider the pH and buffering capacity of topically applied products [40]. The pH of all formulations was found to be in the range of 5.58–6.04 (Table 3). All reported values are close to the pH of the skin and can be safely used topically without causing irritation or any other skin reactions [41].

#### 2.4.2. Viscosity

The viscosities of all bigel formulations were in the range 220.4 to 391.5 cps. An increase in viscosity was observed with an increase in oil concentration, as reported in Table 3.

#### 2.4.3. Spreadability

The spreadability of all bigel formulations was 5.30–6.50 cm (Table 3). Spreadability is a function of the structural viscosity, with lower viscosity meaning better spreadability. The results confirmed a strong correlation between spreadability and viscosity values, signifying good application characteristics [42].

#### 2.4.4. SEM Analysis

Scanning electron microscopic examination was performed to analyze the surface characteristics of the bigel. The micrographs of formulations BG4 and BG8 at 270× and 50× magnification, as shown in Figure 2, revealed the presence of fibrous structures, which could be attributed to the entrapment of the oleogel within the hydrogel phase [43].

### 2.5. In Vivo Studies

#### 2.5.1. Acute Dermal Irritation Studies

Formulations BG4 and BG8 containing the highest concentration of tamanu oil (20%) were selected for the test. The results of dermal irritation studies are shown in Figure 3. No responses, such as erythema, eschar, or edema, were visible on the rabbits after exposure to the bigels for 4 h. Thus, these formulations can be considered non-irritant and safe for topical application [44].

#### 2.5.2. In Vivo Wound-Healing Studies

Based on the results obtained from the evaluation studies, formulations BG1, BG4, and BG8 were selected for the in vivo studies. An excision wound with a diameter of 6 mm was created in all the rats and they were further segregated for treatment. Five groups were treated with pure tamanu oil, commercial formulation (Murivenna^TM^), and test formulations separately, while the control group did not receive any treatment. Treatment was started 24 h post-wound induction and continued daily until complete healing was observed. The wound area was measured on days 3, 6, 9, 12, and 15 of the treatment. The excised wound treated with bigels showed significant wound contraction over a period of 12 days, indicating an accelerated re-epithelialization process compared to the untreated wound (control). The animals did not show any signs of necrosis, inflammation, or hemorrhage and survived throughout the study period. Although the BG4 and BG8 formulations significantly increased wound healing, the best results were observed with BG8 (100% wound contraction on day 12). Overall, the wounds healed and were completely sealed within 15 days post-wound induction, with no evidence of scars [45]. Representative images of wound healing and contraction data are presented in Figure 4, Appendix A and Table 4, respectively.

Epithelialization was observed on the wound area until the eschar had fallen off, without leaving any traces of a raw wound. The period of epithelialization was faster in the BG8 group as the eschar had fallen off on after an average of 6.66 days (Appendix A).

### 2.6. Histopathological Studies

Re-epithelialization and collagen production in full-thickness wounds was assessed by H&E and Masson’s trichome staining. The microscopic examination of all skin specimens revealed successful wound healing in all commercial- and bigel-treated animals (Figure 5 and Appendix A). The control group showed the presence of inflammatory cells, irregular connective tissues, poor collagen deposition, and incomplete epithelial layer formation, in comparison to the treated groups. Tamanu-oil- and murivenna-treated groups showed more collagen fiber deposition, reduced inflammatory cells, and partially developed connective tissues. Bigel-treated groups exhibited normal architecture with a well-developed epidermal layer and an abundance of collagen fibers [46].

## 3. Materials and Methods

### 3.1. Materials

Tamanu oil and Geogard^®^ ECT were purchased from Moksha Lifestyle Products (Delhi, India). Piscean collagen was procured from Himrishi Herbals (Delhi, India) and xanthan gum from Shakthi enterprises (Mumbai, India).

### 3.2. GC-MS Characterization of Tamanu Oil

The chemical composition of tamanu oil was determined using an Agilent 7890B Series gas chromatograph linked with a 5977A Series mass selective detector (Agilent Technologies, Santa Clara, CA, USA). The chromatographic column was HP-5 ms (5% phenyl-methylpolysiloxane; Agilent Technologies, USA) of 30 m length; 0.25 mm internal diameter; and 0.25 µm film thickness. The flow rate of the carrier gas (99.99% pure helium) was maintained at 2.0 mL/min. A 1.0 µL sample (oil diluted with hexane) was injected in split mode (split ratio of 50:1; injector temperature, 290 °C). The oven was programmed at 60 °C as the initial temperature for 2 min, increased to 270 °C at a rate of 4 °C/min and then to 290 °C at a rate of 10 °C/min for a total duration of 65 min. The mass selective detector was operated at an ion source temperature of 270 °C, with electron ionization at 70 eV. The peak areas are represented by the percentage of each compound and their retention times were compared to the calibration curves of the internal standards for compound identification.

### 3.3. In Silico Studies

#### 3.3.1. Preparation of Ligand and Selection of Protein

Ligands selected from the GC-MS report were downloaded in sdf format (3D structures) from PubChem (https://pubchem.ncbi.nlm.nih.gov/, accessed on 16 November 2023) and they were also optimized using the Avogadro tool (https://avogadro.cc/, accessed on 16 November 2023). The structure of the protein target was selected and extracted from Research Collaboratory for Structural Bioinformatics Protein Data Bank (https://www.rcsb.org/, accessed on 16 November 2023). The protein fibronectin (PDB: 5KF4), transforming growth factor beta (TGF-β) (PDB: 5E8W), matrix metalloproteinases (MMPs) (PDB: 3UTZ), tissue inhibitors of metalloproteinases (TIMPs) (PDB: 3V96), platelet-derived growth factor (PDGF) (PDB: 5K5X), vascular endothelial growth factor (VEGF) (PDB: 6T9D), insulin-like growth factor (IGF) (PDB: 4D2R), and various cytokines, such as interleukin-1 (IL-1) (PDB: 7SZL) and IL-6 (PDB: 2L3Y), were selected and downloaded in the pdb format.

#### 3.3.2. Preparation of Protein

Hetero atoms and ligands in the downloaded protein were removed using BIOVIA Discovery Studio (v4.5). The protein target was prepared by the addition of hydrogen atoms and saved in pdb format using Swiss-Pdb Viewer (v4.1).

#### 3.3.3. Molecular Modeling Studies

In order to analyze the binding and molecular interactions of identified active constituents of tamanu oil with therapeutic targets, molecular docking studies were carried out using PyRx with default settings and parameters. The X-ray-resolved crystal structures of the potential were retrieved from the PDB with good resolution and R-free factor. The ligand was uploaded, minimized, and converted to pdbqt format. The molecular docking was performed using PyRX 0.8 software which runs on the Autodock Vina algorithm, the grid box was calculated from the co-crystal ligands, and for pure protein the grid box was generated using the PRANK webserver (https://prankweb.cz/, accessed on 16 November 2023) for carrying out active pocket docking (Appendix A). All complex binding affinity energies were calculated on the basis of ligand conformation at the active binding site, with the root mean square deviation (RMSD) between the original and then the structures taken into account. The number of hydrogen bonds and non-covalent interactions for each complex was calculated using Discovery Studio Visualizer (http://accelrys.com/products/collaborative-science/biovia-discovery-studio/, accessed on 16 November 2023), which produced details, compound images, and interaction images (2D and 3D). We used web-based pkCSM-pharmacokinetic tools to determine the pharmacokinetics (absorption, distribution, metabolism, and excretion); safety; and physicochemical properties of the selected bioactive molecules [43,47].

ADMET profile was analyzed using the pkCSM ADMET descriptors algorithm.

### 3.4. Formulation of Bigels

Micronized xanthan gum was dispersed in 10 mL water for 1h to obtain a gel-like consistency. Geogard^®^ ECT was added along with vanilla fragrance to the above gel to form hydrogel phase. Tamanu oil and Tween 20 were mixed together and heated to 75 °C to obtain an oleogel phase. The oleogel phase was slowly blended with the hydrogel phase to form a bigel.

Eight formulations were prepared containing 5–20% tamanu oil (Appendix A). Formulations BG1–BG4 were prepared with 1% micronized xanthan gum as the hydrogel and formulations BG5–BG8 contained 2% micronized xanthan gum in the hydrogel phase. The concentration of Tween 20 and Geogard^®^ ECT (preservative) was maintained constant at 3% and 1%, respectively, for all formulations. Geogard ECT is a water-soluble, low-odor and low-color, broad spectrum preservative that offers broad spectrum protection in a variety of personal care products. It is a combination of benzyl alcohol, salicylic acid, and sorbic acid. Vanilla fragrance was used to mask the nutty odor of tamanu oil.

### 3.5. Evaluation of Bigels

#### 3.5.1. Organoleptic Evaluation

The bigels were evaluated for their organoleptic properties like color, odor, homogeneity, consistency, phase separation, and texture.

#### 3.5.2. pH

The pH of all formulations was tested using a digital pH meter (MK VI, Systronics, Ahmedabad, India). First, 1g of the bigel was dissolved in 100 mL of water and tested. The measurements were taken in triplicate, and the standard deviation was determined.

#### 3.5.3. Spreadability

A good topical formulation should spread evenly during application. This was evaluated by preparing a thin smear of the bigel on a glass slide with the aid of another glass slide. A weight of 100 g was placed over the glass plate for 5 min to obtain an even smear. The time taken to separate the slides was considered and the spreadability was calculated using the following formula [48].
S = M × L/T
where S is the spreadability, M is the weight applied, L is the length moved by the glass slide during separation and T is the time taken to separate the slides from each other.

#### 3.5.4. Viscosity

The viscosity of all formulations was determined using a Brookfield viscometer (Model DV-II+, Middleboro, MA, USA) with a spindle number of 4 at 10 rpm.

#### 3.5.5. Scanning Electron Microscopic Analysis

The surface characteristics of bigels were observed using scanning electron microscopy (JSM-IT300, Peabody, MA, USA). Dried bigel specimens were coated with a thin layer of gold in an argon atmosphere, glued to aluminum stubs, and observed at 50× and 270× magnification [41].

### 3.6. In Vivo Studies

#### 3.6.1. Acute Dermal Irritation Studies

The likelihood of bigels to cause dermal irritation was investigated on healthy female New Zealand white rabbits in accordance with OECD 404 [40] test guidelines. The experimental protocol was executed after obtaining prior approval from the Institutional Animal Ethical Committee (IAEC Ref No.: XXV/MSRFPH/CEU/M-017/21.10.2021). All the rabbits were acclimatized to the facilities for 2 weeks and individually housed under controlled environmental conditions (20 ± 3 °C/50–60% RH) with a 12 h light and 12 h dark cycle and maintained on a pellet diet and unrestricted supply of drinking water. The fur on the dorsal area of rabbits was carefully shaved 24 h before the test. The animals were distributed into groups of six randomly with one group for control and the others for the formulations. The formulations were applied on the shaved area and secured with gauze and non-irritating tapes. After the exposure period of 4 h, the formulation was removed by rinsing the area with water and the exposed area was closely monitored for any visible change (erythema, redness, and edema). Observations were scored on a scale of 0–4, with 0 indicating the absence of erythema/edema and 4 indicating severe erythema/edema. Scoring was carried out 60 min after removal of formulation, as well as 24, 48, and 72 h later.

#### 3.6.2. In Vivo Wound-Healing Studies

The scar-free wound-healing potential of bigels was determined by an excisional wound model on male albino Wistar rats (weight range of 200–350 g). The fasted rats were weighed and anaesthetized with a mixture of ketamine (80 mg/kg) and xylazine (20 mg/kg) prior to wounding. The dorsal area was depilated and disinfected with 70% ethanol. A full-thickness wound of 6 mm diameter was made using a sterile biopsy punch. The wounded rats were segregated into six groups (*n* = 6) and treatment was carried out with tamanu oil, commercial formulation (Murivenna^TM^), and test formulations BG1, BG4, and BG8 while the control group did not receive any treatment. On the 3rd, 6th, 9th, 12th, and 15th day, the wound area was measured using a scale and the % closure calculated [45].
$$\%\mathrm{Wound}\mathrm{closure}=\frac{\mathrm{Initial}\mathrm{wound}\mathrm{area}-\mathrm{Specific}\mathrm{day}\mathrm{wound}\mathrm{area}}{\mathrm{Initial}\mathrm{wound}\mathrm{area}}\times 100$$

The number of days from the day of wound induction till the falling off of eschar without leaving any raw wound behind was considered to be the period of epithelialization.

### 3.7. Histopathological Studies

At the end of the 15-day treatment period, one animal from each group was anaesthetized with a mixture of ketamine and xylazine (80 mg/kg and 20 mg/kg, i.p., respectively) and sacrificed. Restored skin samples from each group were carefully isolated and stored in 10% *v*/*v* formalin solution. The specimens were embedded in paraffin, divided into sections of 4 μm thickness, and stained with hematoxylin and eosin (H&E) and Masson’s trichrome for microscopic evaluation of epithelization, fibroblast proliferation, keratinization, and collagen formation [49].

### 3.8. Statistical Analysis

The wound contraction data were statistically analyzed by one-way ANOVA followed by comparisons with the control group using Dunnett’s test. GraphPad Instat 3.1 software was employed for statistical interpretations. Values with *p* > 0.05 were considered not significant (ns), *p* < 0.05 moderately significant (*), and *p* < 0.01 as highly significant (**).

## 4. Conclusions

The main objective of the present study was to develop bigels of tamanu oil and assess its anti-scarring activity. Bigels were prepared with 5 to 20% tamanu oil and 1–2% micronized xanthan gum. In silico studies were performed and it was found that the components calanolide A, inophyllum C, and 4-norlanosta-17(20),24-diene-11,16-diol-21-oic acid, 3-oxo-16,21-lactone had a docking score of −9.8 kcal/mol, −11.3 kcal/mol, and −11.1 kcal/mol, respectively, with 5E8W, a cytokine TGF-β1 receptor. The standard staurosporine reported a docking score of −8.3 kcal/mol. Acute dermal irritation performed on rabbits showed no irritation, erythema, eschar, or edema. In vivo wound-healing studies were performed to compare the effectiveness of bigel formulations and standard (Murivenna oil) in healing and scar reduction. Based on the results obtained, formulation BG8 was found to be better compared to all other formulations and the standard as it was able to heal the wound within 12 days without leaving behind a scar. Thus, the present study concludes that bigels of tamanu oil are a promising topical product with good scar-healing activity.

## Figures and Tables

**Figure 1 pharmaceuticals-17-00102-f001:**
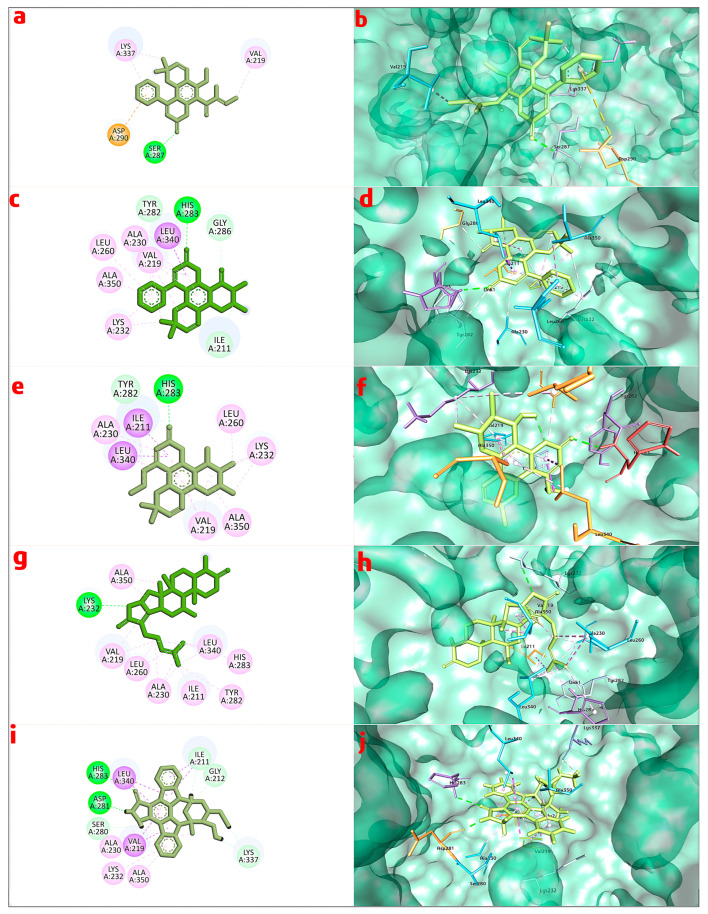
Binding interaction of calophyllolide with 5E8W, (**a**) 2D and (**b**) 3D interaction; binding interaction of inophyllum C, (**c**) 2D and (**d**) 3D interaction; binding interaction of calanolide A with 5E8W, (**e**) 2D and (**f**) 3D interaction; binding interaction of norlanosta-17(20),24-diene-11,16-diol-21-oic acid, 3-oxo-16,21-lactone ring with 5E8W, (**g**) 2D and (**h**) 3D interaction; binding interaction of staurosporine with 5E8W, (**i**) 2D and (**j**) 3D interaction.

**Figure 2 pharmaceuticals-17-00102-f002:**
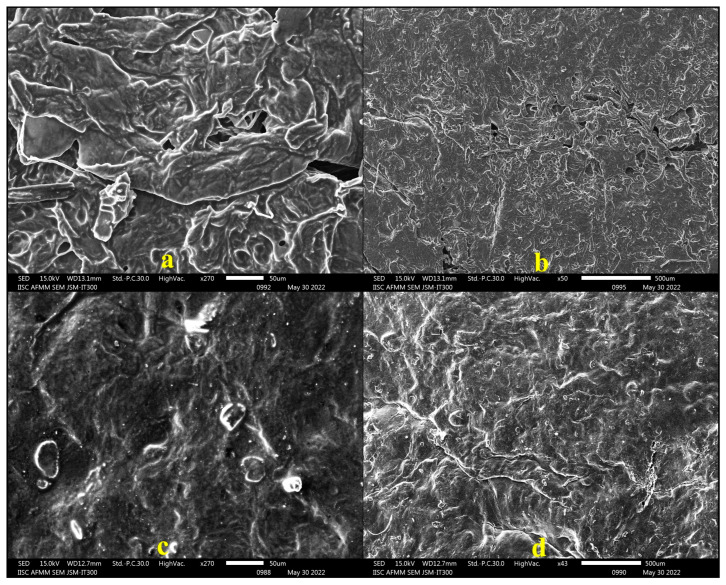
SEM micrographs of (**a**) BG4 at 270×, (**b**) BG4 at 50×, (**c**) BG8 at 270×, (**d**) BG8 at 50×.

**Figure 3 pharmaceuticals-17-00102-f003:**
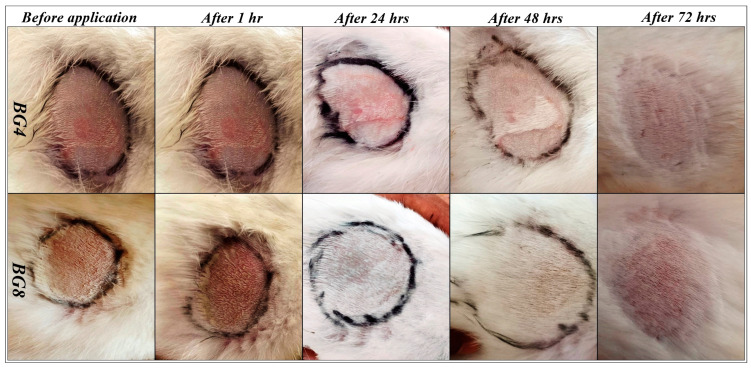
Photographs of dermal irritation studies carried out on rabbits.

**Figure 4 pharmaceuticals-17-00102-f004:**
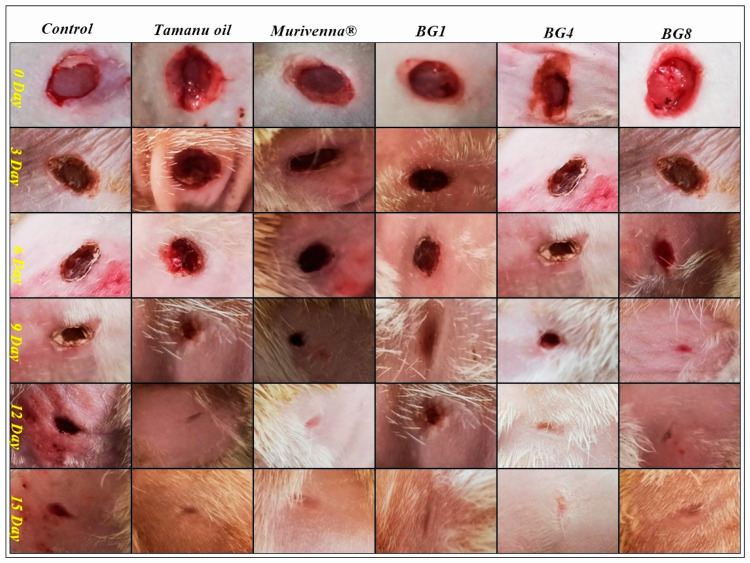
Representative images of wound control and wounds treated with standard and bigels for 15 days in rats.

**Figure 5 pharmaceuticals-17-00102-f005:**
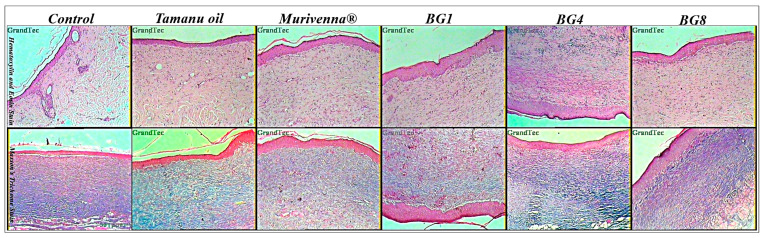
Photomicrographs (100×) showing H&E stain and MT stain section of skin tissues at day 15 for control, standard-, and formulation-treated groups of rats.

**Table 1 pharmaceuticals-17-00102-t001:** Components identified in GC-MS analysis of tamanu oil.

Sl.No.	Compounds	Component RT (Min)	SMILES	Percentage
1	Calanolide A	31.03	CCCC1=CC(=O)OC2=C1C3=C(C=CC(O3)(C)C)C4=C2[C@H]([C@@H]([C@H](O4)C)C)O	2.02
2	Calophyllolide	33.52	CC=C(C)C(=O)C1=C(C2=C(C3=C1OC(=O)C=C3C4=CC=CC=C4)OC(C=C2)(C)C)OC	1.92
3	Inophyllum C	24.91	C[C@@H]1[C@H](OC2=C(C1=O)C3=C(C(=CC(=O)O3)C4=CC=CC=C4)C5=C2C=CC(O5)(C)C)C	3.65
4	Oleic acid	26.16	CCCCCCCC/C=C\CCCCCCCC(=O)O	0.85
5	Linoleic acid	26.16	CCCCC/C=C\C/C=C\CCCCCCCC(=O)O	0.64
6	Palmitic acid	23.27	CCCCCCCCCCCCCCCC(=O)O	0.23
7	Stearic acid	26.38	CCCCCCCCCCCCCCCCCC(=O)O	0.98
8	4-Norlanosta-17(20),24-diene-11,16-diol-21-oic acid, 3-oxo-16,21-lactone	29.66	CC1C2CCC3(C(C2(CCC1=O)C)C(CC4C3(CC5C4=C(C(=O)O5)CCC=C(C)C)C)O)C	2.3
9	Pentacosanoic acid	25.28	CCCCCCCCCCCCCCCCCCCCCCCCC(=O)O	0.66

**Table 2 pharmaceuticals-17-00102-t002:** Binding affinities of constituents of tamanu oil with TGF-β type 1 kinase (5E8W).

Compounds	Docking Score kcal/mol	H-Bond Interactions		Hydrophobic Interactions	External Bond Interactions
		Residue	Distance (Å)		
Calophyllolide	−8.6	Ser287	2.3	Val219, Lys337	-
Inophyllum C	−11.3	His283	2.47	Val219, Ala230, Lys232, Leu260, Leu340, Ala350, Asp351	-
Calanolide A	−9.8	Ser280, His283	2.88 2.31	Ile211, Val219, Lys232, Leu260, Leu340	-
Oleic acid	−5.3	Ser280	2.56	Ile211, Val219, Leu260, Leu340, Ala350, Asp351	-
Linoleic acid	−6.4	Ser280, His283	3.25 1.87	Ile211, Val219, Lys232, Tyr249, Leu260, Phe262, Asp351	-
Palmitic acid	−5.9	His283	2.9	Lys232, Tyr249, Phe262, Leu278, Leu340, Lys232	-
4-Norlanosta-17(20),24-diene-11,16-diol-21-oic acid, 3-oxo-16,21-lactone	−11.1	Ser287	3.08	Ile211, Val219, Ala230, Leu260, Tyr282, Lys337, Leu340, Ala350, Asp351	Lys232 (salt bridge)
Hyenic acid	−5	Val231	3.29	Ile211, Val219, Lys232, Leu260, Lys337, Leu340, Ala350, Asp351	
Staurosporine	−8.6	Ser280, Asp281, His 283	2.57 2.76 1.94	Ile 211, Gly212, Lys337	Val219, Ala230, Lys232, Leu340, Ala350 π stacking

**Table 3 pharmaceuticals-17-00102-t003:** Evaluation of bigels.

Parameters	BG1	BG2	BG3	BG4	BG5	BG6	BG7	BG8
pH	5.82 ± 0.05	5.96 ± 0.06	5.92 ± 0.12	6.04 ± 0.07	5.96 ± 0.17	5.58 ± 0.03	5.62 ± 0.13	5.76 ± 0.10
Spreadability (cm)	6.50 ± 0.36	6.10 ± 0.26	5.93 ± 0.25	5.63 ± 0.30	5.46 ± 0.40	5.36 ± 0.05	5.30 ± 0.10	5.26 ± 0.15
Viscosity (cps)	220.4 ± 0.96	238.9 ± 0.85	252.9 ± 0.65	273.8 ± 0.05	313.0 ± 0.15	337.4 ± 0.25	378.4 ± 0.05	391.5 ± 0.28

**Table 4 pharmaceuticals-17-00102-t004:** Statistical analysis of wound contraction.

Group	Control	Tamanu Oil	Murivenna	BG1	BG4	BG8
Day 3	0.573 ± 0.0049	0.538 ± 0.0047 **	0.555 ± 0.0042 ^ns^	0.561 ± 0.0094 ^ns^	0.511 ± 0.0060 **	0.505 ± 0.0042 **
Day 6	0.470 ± 0.0057	0.460 ± 0.0051 ^ns^	0.451 ± 0.0047 *	0.463 ± 0.0042 ^ns^	0.415 ± 0.00428 **	0.405 ± 0.0042 **
Day 9	0.418 ± 0.0094	0.350 ± 0.0036 **	0.358 ± 0.0095 **	0.380 ± 0.0057 **	0.273 ± 0.0066 **	0.248 ± 0.0095 **
Day 12	0.251 ± 0.0104	0.112 ± 0.0107 **	0.098 ± 0.0047 **	0.1283 ± 0.0060 **	0.0833 ± 0.0066 **	0.00 ± 0.00 **
Day 15	0.083 ± 0.0025	0.00 ± 0.00 **	0.00 ± 0.00 **	0.00 ± 0.00 **	0.00 ± 0.00 **	0.00 ± 0.00 **

All values represent the diameter of the wound (in cm) on different days of measurement and are expressed as mean ± SEM (*n* = 6); one-way ANOVA followed by Dunnett’s test where ns *p* > 0.05, * *p* < 0.05, ** *p* < 0.01, in comparison to control. (ns = non-significant, * moderately significant, ** highly significant).

## Data Availability

Data are available in the article and the Appendix A.

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
