# Peer review of "An Integrated Computational and Experimental Approach to Formulate Tamanu Oil Bigels as Anti-Scarring Agent"

_pharmaceuticals, 2024, doi:10.3390/ph17010102_

Round 1

Reviewer 1 Report (Previous Reviewer 2)

Comments and Suggestions for Authors

As I wrote in my earlier comments (on the previous version of the manuscript) my main objection was to the choice of reference drug (bleomycin) and docking method. The rest of the manuscript was ok with me. This version of the manuscript differs little from the earlier version. Nevertheless, the authors removed the docking for bleomycin. Staurosporine was chosen. Of course, it would have been better, for example, Decorin, S58 aptamer or Lerdelimumab. But ok, Staurosporine as a reference is acceptable.

The authors also added the percentage of ingredients in the mixture, which is what I asked for.

So, only one comment remains. Such as I previously wrote,

In section 3.3.1, the authors wrote that the ligand structures were taken in sdf format. Unfortunately, these structures are completely flat, 2D, and are not suitable for docking. Can the authors explain and describe how they optimized the ligand structures? If they did not do this and used flat structures, 2D, the docking performed is incorrect and the results unreliable.

Besides, in section 3.3.1 they list several crystal structures, lines 298-304. Why? Only docking for the 5E8W structure is included in the paper. 

Author Response

As I wrote in my earlier comments (on the previous version of the manuscript) my main objection was to the choice of reference drug (bleomycin) and docking method. The rest of the manuscript was ok with me. This version of the manuscript differs little from the earlier version. Nevertheless, the authors removed the docking for bleomycin. Staurosporine was chosen. Of course, it would have been better, for example, Decorin, S58 aptamer or Lerdelimumab. But ok, Staurosporine as a reference is acceptable. The authors also added the percentage of ingredients in the mixture, which is what I asked for. So, only one comment remains. Such as I previously wrote,        In section 3.3.1, the authors wrote that the ligand structures were taken in sdf format. Unfortunately, these structures are completely flat, 2D, and are not suitable for docking. Can the authors explain and describe how they optimized the ligand structures? If they did not do this and used flat structures, 2D, the docking performed is incorrect and the results unreliable.

We thank the reviewer for the valuable comments to improve our manuscript, we have utilized 3D form of ligand structures which was optimized by Avogadro tool for carrying out Molecular Docking and the same was incorporated in the updated manuscript.

Besides, in section 3.3.1 they list several crystal structures, lines 298-304. Why? Only docking for the 5E8W structure is included in the paper. 

We thank the reviewer for the valuable comments, we have incorporated the Docking results of other targets in our supplementary data, as 5E8W this was found be better target (binding score and binding interaction) than the other targets we have elaborated and explained in detail in the manuscript.

Reviewer 2 Report (New Reviewer)

Comments and Suggestions for Authors

In the manuscript  the authors investigated the potential of Tamanu oil as an active ingredient for the development of bigels with significant scar-reducing properties. An in silico study elucidating its interaction with 5E8W (TGF-β receptor type-1), a membrane- bound TGF beta receptor protein, was performed to demonstrate its clinical potential.

However, the authors should correct manuscript according to following suggestions:

a) The Figure 1 should be corrected because  images b, d, f, h, j signatures of amino acids is unreadable. The grey color of protein should be changed. The figure with structure of TGF-β type 1 kinase (5E8W) with shown binding site would be included in the manuscript.

b) The caption to figure 1 should be also corrected.

c) The units of docking score should be given in the manuscript.

d) In the table 2 the hydrophobic bond should be changed to hydrophobic interactions.

e) The structures of ingredients of Tamanu oil should be presented in supplementary file.

f) Do all active substances in Tamanu oil will interact with one protein?

 g) The disscusion of the aminoacids important for binding compounds in the selected protein should be included.

Author Response

In the manuscript the authors investigated the potential of Tamanu oil as an active ingredient for the development of bigels with significant scar-reducing properties. An in-silico study elucidating its interaction with 5E8W (TGF-β receptor type-1), a membrane- bound TGF beta receptor protein, was performed to demonstrate its clinical potential.

However, the authors should correct manuscript according to following suggestions:

  1. a) The Figure 1 should be corrected because images b, d, f, h, j signatures of amino acids is unreadable. The grey color of protein should be changed. The figure with structure of TGF-β type 1 kinase (5E8W) with shown binding site would be included in the manuscript.

We thank the reviewer for the valuable comments to improve our manuscript, we increased the size of Amino acid name in 3d images (3d Semi Bold – size 20 in discovery, the maximum size) and but while collating the images, size got compressed, kindly accept the present form of image.

  1. b) The caption to figure 1 should be also corrected.

We thank the reviewer for the valuable comments, we have modified the same and updated

  1. c) The units of docking score should be given in the manuscript.

We thank the reviewer for the valuable comments, we have modified the same and updated in the manuscript.

  1. d) In the table 2 the hydrophobic bond should be changed to hydrophobic interactions.

We thank the reviewer for the valuable comments, we have modified the same and updated in the manuscript.

  1. e) The structures of ingredients of Tamanu oil should be presented in supplementary file.

We thank the reviewer for the valuable comments, we have modified the same and updated in the supplementary table.

  1. f) Do all active substances in Tamanu oil will interact with one protein?

Thanks for the valuable comment. Initially we tried to explore the possible synergetic mechanism of Tamanu oil with the various protein targets which plays the major role in the scar formation such as Collagen Type I, Fibronectin, Elastin, TGF-beta (Transforming Growth Factor-beta), MMPs (Matrix Metalloproteinases), TIMPs (Tissue Inhibitors of Metalloproteinases), PDGF (Platelet-Derived Growth Factor), VEGF (Vascular Endothelial Growth Factor), IGF (Insulin-like Growth Factor) and Various cytokines, such as IL-1 (Interleukin-1) and IL-6. Binding energy for the selected targets was found to be less than -7.7 Kcal/mol except TGF target, hence we have selected TGF as the key target of tamanu oil key constituents in producing anti scar properties. Not only based on binding score, based on binding interaction also we have selected TGF target. The same we reported in our manuscript.

  1. g) The disscusion of the aminoacids important for binding compounds in the selected protein should be included.

We thank the reviewer for the valuable comments, we have modified the same and updated in the manuscript.

Reviewer 3 Report (New Reviewer)

Comments and Suggestions for Authors

The article presents a comprehensive study of tamanu oil as an anti-scarring agent.

·         Table 1: What does mean “Percentage”? Is it volume fraction, or mass fraction?

·         Table 2: Missing units for binding affinity.

·         Figure 1: Symbols for amino acid residues are invisible at the presentation of 3D interactions (Figures b, d, f, h, j)

·         Missing deeper discussion for the sections: 2.2. In silico studies -Molecular modeling;

There are lot of spelling mistakes through the manuscript and supplementary information file:

·         symbol for kilocalorie is kcal, not Kcal

·         line 23-25, 103, 143, 145, 149, 155, 394: The names of compounds should start with small letters.

·         line 27, 91, 100, 124: Tamanu oil should start with a small letter, as well as Murivenna oil (line 31).

·         line 63: Tamanu tree should start with small letter.

·         line 64, 75, 78: Latin names (Calophyllum inophyllum) should be written italics.

·         line 126: Should be written: collagen type I (with small letters)

·         line 137, 140, 145: Staurosporine should start with a small letter

Author Response

The article presents a comprehensive study of tamanu oil as an anti-scarring agent.

  • Table 1: What does mean “Percentage”? Is it volume fraction, or mass fraction?

We thank the reviewer for the valuable comments, its volume fraction. The same was written in our manuscript Section 3.2.

  • Table 2: Missing units for binding affinity.

We thank the reviewer for the valuable comments, we have modified the same and updated in the manuscript.

  • Figure 1: Symbols for amino acid residues are invisible at the presentation of 3D interactions (Figures b, d, f, h, j)

We thank the reviewer for the valuable comments to improve our manuscript, we increased the size of Amino acid name in 3d images (3d Semi Bold – size 20 in discovery, the maximum size) and but while collating the images, size got compressed, kindly accept the present form of image.

  • Missing deeper discussion for the sections: 2.2. In silico studies -Molecular modeling;

We thank the reviewer for the valuable comments, we have modified the same and updated in the manuscript.

There are lot of spelling mistakes through the manuscript and supplementary information file:

  • symbol for kilocalorie is kcal, not Kcal
  • line 23-25, 103, 143, 145, 149, 155, 394: The names of compounds should start with small letters.
  • line 27, 91, 100, 124: Tamanu oil should start with a small letter, as well as Murivenna oil (line 31).
  • line 63: Tamanu tree should start with small letter.
  • line 64, 75, 78: Latin names (Calophyllum inophyllum) should be written italics.
  • line 126: Should be written: collagen type I (with small letters)
  • line 137, 140, 145: Staurosporine should start with a small letter

We thank the reviewer for the valuable comments, we have modified the same and updated in the manuscript.

Round 2

Reviewer 1 Report (Previous Reviewer 2)

Comments and Suggestions for Authors

The authors responded to my comment. I have no further comments.

This manuscript is a resubmission of an earlier submission. The following is a list of the peer review reports and author responses from that submission.

Round 1

Reviewer 1 Report

Comments and Suggestions for Authors

This work conducted by Krishnappa et al explored the possibility of making bigels out of tamanu oil for anti-scarring purpose. The authors employed the GS-MS technique to analyze the active compounds in the tamanu oil, used molecular docking method to analyze the binding modes of the above-mentioned compounds bound to TGF- β1 receptor. The authors also did in vivo studies on rabbits and rats. This work in general is interesting and a few things need to be further considered. 

1) In the GC-MS results shown in Table 1, it would be better to include the percentage of each compounds and replace the 'Structure' column with SMILES string for each compound.

2) The molecular docking studies carried out by authors is based on the assumption that TGF- β1 receptor is the sole receptor. If so, the authors need to provide more proof to support this assumption. 

3) For figures 4 and 5, the authors need to indicate whether it is from rabbit or rat. 

Author Response

This work conducted by Krishnappa et al explored the possibility of making bigels out of tamanu oil for anti-scarring purpose. The authors employed the GS-MS technique to analyze the active compounds in the tamanu oil, used molecular docking method to analyze the binding modes of the above-mentioned compounds bound to TGF- β1 receptor. The authors also did in vivo studies on rabbits and rats. This work in general is interesting and a few things need to be further considered. 

1) In the GC-MS results shown in Table 1, it would be better to include the percentage of each compound and replace the 'Structure' column with SMILES string for each compound.

Answer: We thank the reviewer for the valuable comments to improve our manuscript, now we have incorporated the same in the updated manuscript.

2) The molecular docking studies carried out by authors is based on the assumption that TGF- β1 receptor is the sole receptor. If so, the authors need to provide more proof to support this assumption. 

Answer: Thanks for your valuable comment. In our future work, we will carry out molecular dynamics study and other QM/MM studies. Due to infrastructure constraints, we couldn’t perform the above said studies. Kindly consider the present works.

3) For figures 4 and 5, the authors need to indicate whether it is from rabbit or rat. 

Answer: We thank the reviewer for the valuable comments to improve our manuscript, now we have incorporated the same in the updated manuscript.

Reviewer 2 Report

Comments and Suggestions for Authors

In this manuscript, the authors present results anti-scarring properties of tamanu oil. In vivo and in silico methods were applied. The paper is appropriately written. The background is adequate, and the experiment is correctly done. I have a few comments:

Molecular docking is poorly described. The authors limit themselves only to providing binding affinity values and a table of interactions. They do not discuss it in any way. The fact that the binding affinity is negative (by the way, what is the docking score unit? J, calorie?), shows only that there is the possibility of forming a complex. What is important is whether the molecules interact with the amino acids that are appropriate from the point of view of anti-scarring properties. The authors should first describe exactly how the molecule should interact and then show that the study compounds interact in a similar way.

Why did the authors choose bleomycin as the standard, as the reference drug? It is a compound with anti-tumour activity, and the target of action is DNA. The authors should have chosen a drug that has anti-scarring activity and whose target is TGF- β1 receptor. I don’t know, maybe Decorin, S58 aptamer or Lerdelimumab? In that case, docking would make sense. Now, not so much.

The authors list several components of the Tamanu oil. Since it is not known which ingredient or ingredients cause anti-scarring properties, it would be good to give the percentage of each ingredient in the mixture.

Author Response

In this manuscript, the authors present results anti-scarring properties of tamanu oil. In vivo and in silico methods were applied. The paper is appropriately written. The background is adequate, and the experiment is correctly done. I have a few comments:

  1. Molecular docking is poorly described. The authors limit themselves only to providing binding affinity values and a table of interactions. They do not discuss it in any way. The fact that the binding affinity is negative (by the way, what is the docking score unit? J, calorie?), shows only that there is the possibility of forming a complex. What is important is whether the molecules interact with the amino acids that are appropriate from the point of view of anti-scarring properties. The authors should first describe exactly how the molecule should interact and then show that the study compounds interact in a similar way.

Answer: We thank the reviewer for the valuable comments to improve our manuscript, now we have incorporated the same in the updated manuscript.

  1. Why did the authors choose bleomycin as the standard, as the reference drug? It is a compound with anti-tumour activity, and the target of action is DNA. The authors should have chosen a drug that has anti-scarring activity and whose target is TGF- β1 receptor. I don’t know, maybe Decorin, S58 aptamer or Lerdelimumab? In that case, docking would make sense. Now, not so much.

Answer: Thanks for the valuable comment. As the above-mentioned drugs are Macromolecular drugs, they cannot be used for carrying out small molecule protein docking. Based on the literatures we found bleomycin is used as anti-scar agents (https://doi.org/10.1533/9780857093301.1.77; https://www.nature.com/articles/bjc1993100 ). Hence, we have selected the same as standard.

  1. The authors list several components of the Tamanu oil. Since it is not known which ingredient or ingredients cause anti-scarring properties, it would be good to give the percentage of each

Answer: Thank you. It's important to note that the exact percentage of each component in Tamanu oil can vary depending on the source and processing method, hence we couldn’t give the exact percentage of active ingredients. We tried to explore which are the key active ingredient for ant-scar property through molecular docking, we found as multiple compounds showed better inhibition, so concluded as the synergistic effect of all, it produced better effect in in vivo studies.

Reviewer 3 Report

Comments and Suggestions for Authors

The manuscript entitled "An Integrated Computational and Experimental Approach to Formulate Tamanu Oil Bigels as Anti-Scarring Agent" reports an interesting study about Tamanu Oil. The manuscript may be considered for publication subject to the rectification of following issues:

1. The rationality of selecting TGF-β for molecular docking study is not clear. Is there any experimental evidence related to the constituents of Tamanu Oil against the selected target? If not then there are several proteins/cascades which are involved in scar and why this particular target was selected??

2. Table 2: Residues column is not required; rather interacting residues can be presented as "Val219" (for example, first row in the Table). Authors may refer to other papers for presenting docking data. It should include H-bond interaction also. Types of hydrophobic interactions must also be specified.

3. Presentation of docking images must be improved. Also, only best compound and standard drug can be presented and rest of them should be shifted to the supplementary information.

Comments on the Quality of English Language

Minor issues.

Author Response

The manuscript entitled "An Integrated Computational and Experimental Approach to Formulate Tamanu Oil Bigels as Anti-Scarring Agent" reports an interesting study about Tamanu Oil. The manuscript may be considered for publication subject to the rectification of following issues:

  1. The rationality of selecting TGF-β for molecular docking study is not clear. Is there any experimental evidence related to the constituents of Tamanu Oil against the selected target? If not then there are several proteins/cascades which are involved in scar and why this particular target was selected??

Answer: Thank you for valuable comment. The well-known cytokine TGF-β is essential for tissue repair and wound healing, including the creation of scars. It controls a number of biological functions, including the synthesis of extracellular matrix, differentiation, and proliferation of cells all of which are critical for the development and remodeling of scar tissue. Thus, focusing on TGF-β can offer information about possible treatments to control the formation of scars.

  1. Table 2: Residues column is not required; rather interacting residues can be presented as "Val219" (for example, first row in the Table). Authors may refer to other papers for presenting docking data. It should include H-bond interaction also. Types of hydrophobic interactions must also be specified.

Answer: We thank the reviewer for the valuable comments to improve our manuscript, now we have incorporated the same in the updated manuscript.

  1. Presentation of docking images must be improved. Also, only best compound and standard drug can be presented and rest of them should be shifted to the supplementary information.

Answer: Thank you for the valuable comment. Among the selected eight compounds from Tamanu oil, we kept four compound images with the standard as their scores are equivalent to selected standard.

Reviewer 4 Report

Comments and Suggestions for Authors

attached file

Comments on the Quality of English Language

Author Response

I would like to congratulate the authors on their revised version of the manuscript entitled “An Integrated Computational and Experimental Approach to Formulate Tamanu Oil Bigels as Anti-Scarring Agent” (Manuscript ID: pharmaceuticals-2583349). In this study, in silico and in vivo studies to avail tamanu oil into treatment of skin problems.

Overall, this is an interesting study, however, I have spotted some weaknesses in the manuscript which should be clarified before publication in the Pharmaceuticals.

Major Points

1) The abstract should clarify how computational and experimental analysis are related.

Answer: We thank the reviewer for the valuable comments to improve our manuscript, now we have incorporated the same in the updated manuscript.

2) In the In silico studies: (i) How docking procedures were validated (e.g. re-docking)?; (ii) Does PyRx 0.8 run docking simulations? I believe it has been used as a tool to support a true docking program; (iii) docking parameters should be added to the manuscript (e.g. search algorithm, scoring function, docking program, coordinates…); (iv) To add the proper reference for 5E8W PDB (https://doi.org/10.1107/S2059798316003624).

Answer: Thank you for the valuable comment. (i) Already we have optimized our docking procedure by carrying out Redocking and Cross-Docking Studies (Few of publications for your reference https://doi.org/10.3390/molecules28114541; https://doi.org/10.1080/07391102.2023.2246562; https://doi.org/10.1142/S273741652350045X )

(ii) PyRx 0.8 is the tool for running docking studies based on Autodock vina algorithm; (iii) & (iv) We have updated the same in our updated manuscript

3) To improve the discussion for computational merged into experimental analysis.

Answer: We thank the reviewer for the valuable comments to improve our manuscript, now we have incorporated the same in the updated manuscript.

Minor Points

1) The quality (resolution and description) of the Figures should be improved.

Answer: Thanks for the valuable comments. We have increased the resolution to 600dpi

2) To add the proper reference for “BIOVIA Discovery studio software”

Answer: Thanks for the valuable comments. We have updated the same in manuscript.

3) I suggest an English revision of the whole manuscript (for some “typos”).

Answer: Thanks for the valuable comments. We have updated the same in manuscript.

Round 2

Reviewer 2 Report

Comments and Suggestions for Authors

Ad2. Of course, I am not disputing that bleomycin can have neither scarring properties. Just that its molecular target is not necessarily TGF. Maybe it does, maybe it doesn't. Therefore, the authors should use a drug that acts on TGF as the reference drug. Such as I mentioned in my comment. And then everything would be ok.

Of course, I guessed the real reason for this choice. The authors' response confirmed this. That's right, autodock vina or autodock is not the right program to dock these drugs They should use programs like HPEPDOCK or HDDOCK for docking.

Using autodock vina (or autodock) for docking Bleomycin is also problematic. AutoDockVina software is mainly dedicated to the interaction of small molecules. Bleomycin is just a big molecule. And a very flexible one at that. I know, I know, there are papers where bleomycin was docked using autodock vina. Unfortunately, autodock has a limitation on the number of rotatable bonds. The limit in AutoDock Vina is 32. I think for bleomycin this number is higher. In addition, the authors, as they wrote, used pubchem sdf files for docking, which are completely flat. Which is very inappropriate. There should be 3D structures.

To sum up - the way the authors did the docking is controversial and, in my opinion, the results are unreliable. What should they have done? Two choices: either do the docking again (especially for the reference drug), using not autodock vina but It would  use software for peptide-protein or protein-protein interactions, with a global docking method and flexible peptide and protein, and with 3D structure ligands. Or (if they can't do it) remove docking from this work. In my opinion, this work without docking is ok. And sufficient for publication.

Ad.3 That is exactly what I am asking, What is the percentage composition of the  tamanu oil, that the authors studied, used? Since they used chromatography and mass spectroscopy they can calculate these values. Especially, since in lines 292-293 they wrote about it themselves: "The peak areas are represented by the percentage of each compound".

Author Response

Ad2. Of course, I am not disputing that bleomycin can have neither scarring properties. Just that its molecular target is not necessarily TGF. Maybe it does, maybe it doesn't. Therefore, the authors should use a drug that acts on TGF as the reference drug. Such as I mentioned in my comment. And then everything would be ok.

Of course, I guessed the real reason for this choice. The authors' response confirmed this. That's right, autodock vina or autodock is not the right program to dock these drugs They should use programs like HPEPDOCK or HDDOCK for docking.

Using autodock vina (or autodock) for docking Bleomycin is also problematic. AutoDockVina software is mainly dedicated to the interaction of small molecules. Bleomycin is just a big molecule. And a very flexible one at that. I know, I know, there are papers where bleomycin was docked using autodock vina. Unfortunately, autodock has a limitation on the number of rotatable bonds. The limit in AutoDock Vina is 32. I think for bleomycin this number is higher. In addition, the authors, as they wrote, used pubchem sdf files for docking, which are completely flat. Which is very inappropriate. There should be 3D structures.

To sum up - the way the authors did the docking is controversial and, in my opinion, the results are unreliable. What should they have done? Two choices: either do the docking again (especially for the reference drug), using not autodock vina but It would use software for peptide-protein or protein-protein interactions, with a global docking method and flexible peptide and protein, and with 3D structure ligands. Or (if they can't do it) remove docking from this work. In my opinion, this work without docking is ok. And sufficient for publication.

Answer: We thank the reviewer for the valuable comments to improve our manuscript, now we have updated manuscript with the right standard drug Staurosporine, which is the cocrystal bounded with the selected protein TGF (PDB:5E8W). Now the selected test and standard molecules are small molecules, so structure-based drug design method by using Autodock Vina / Autodock tools will be the suitable for carrying out docking studies. Hence, we haven’t tried for protein-protein docking studies. Few of our publication based the above techniques are for you reference https://doi.org/10.3390/molecules28114541; https://doi.org/10.1080/07391102.2023.2246562; https://doi.org/10.1142/S273741652350045X, where we used Autodock Vina for carrying out Structure based drug design. Kindly consider our justification.

Ad.3 That is exactly what I am asking, What is the percentage composition of the tamanu oil, that the authors studied, used? Since they used chromatography and mass spectroscopy, they can calculate these values. Especially, since in lines 292-293 they wrote about it themselves: "The peak areas are represented by the percentage of each compound".

Answer: We thank the reviewer for the valuable comments to improve our manuscript, now we have incorporated the percentage of active constituents of tamanu oil in the updated manuscript table 1.

Reviewer 3 Report

Comments and Suggestions for Authors

The comments were not addressed properly. Previous comment is being repeated: "Is there any experimental evidence related to the constituents of Tamanu Oil against the selected target? If not then there are several proteins/cascades which are involved in scar and why this particular target was selected?" Author should provide experimental evidence against the selected target otherwise use several targets and suggest the best probable target based on molecular docking. The legibility of Figure 1 must be improved. What is to be interpreted from the images presented in right panel??? Table 2 must be revised in a meaningful way (refer Table 3 in http://dx.doi.org/10.7324/JAPS.2023.117478).

Comments on the Quality of English Language

Minor editing of English language required

Author Response

The comments were not addressed properly. Previous comment is being repeated: "Is there any experimental evidence related to the constituents of Tamanu Oil against the selected target? If not, then there are several proteins/cascades which are involved in scar and why this particular target was selected?" Author should provide experimental evidence against the selected target otherwise use several targets and suggest the best probable target based on molecular docking.

Answer: Thanks for the valuable comment. Initially we tried to explore the possible synergetic mechanism of Tamanu oil with the various protein targets which plays the major role in the scar formation such as Collagen Type I, Fibronectin, Elastin, TGF-beta (Transforming Growth Factor-beta), MMPs (Matrix Metalloproteinases), TIMPs (Tissue Inhibitors of Metalloproteinases), PDGF (Platelet-Derived Growth Factor), VEGF (Vascular Endothelial Growth Factor), IGF (Insulin-like Growth Factor) and Various cytokines, such as IL-1 (Interleukin-1) and IL-6. Binding energy for the selected targets was found to be less than -7.7 Kcal/mol except TGF target, hence we have selected TGF as the key target of tamanu oil key constituents in producing anti scar properties. The same we reported in our manuscript.

The legibility of Figure 1 must be improved.

Answer: Thanks for the valuable comment. We have improvised the figure quality to 1200 dpi.

What is to be interpreted from the images presented in right panel???

Answer: Thanks for the valuable comment. Protein ligand complex interaction as 2 dimensional on left panel and the same complex as 3 dimensional in the right panel.

Table 2 must be revised in a meaningful way (refer Table 3 in http://dx.doi.org/10.7324/JAPS.2023.117478).

Answer: Thanks for the valuable comment. We have modified the table as per the reviver instruction.

Reviewer 4 Report

Comments and Suggestions for Authors

Dear Editor,

In this revised version of the manuscript, I am pleased to note that the authors have addressed my prior concerns. I would like to make one final recommendation concerning the consistent unit employed for expressing binding affinity. While "Kcal/mol" is employed in the abstract, "KJ/mol" is used in the section labeled "2.2 In silico studies." I hold a strong conviction that "Kcal/mol" is the appropriate unit of measurement in this context.

Upon reviewing and considering these points, I confidently recommend this manuscript for publication.

Comments on the Quality of English Language

Minor editing of English language required

Author Response

Answer: We thank the reviewer for the valuable comments to improve our manuscript, now we have incorporated the same in the updated manuscript.

Round 3

Reviewer 2 Report

Comments and Suggestions for Authors

Ok. I have no more comments.

Author Response

We thank the reviewer for the valuable comments to improve our manuscript.

Reviewer 3 Report

Comments and Suggestions for Authors

Largely none responsive to previous comments, the manuscript cannot be recommended for publication in its current state. 1) Improving legibility is not just increasing the dpi but the fonts should be clearly visible and the meaningful presentation; 2) Regarding the target selection, as replied by the authors, the results should be presented (in manuscript and supplementary information) and properly discussed; 3) Showing similar pocket repeatedly in surface rendering is meaningless; 4) Table 2 was not modified as suggested.

Comments on the Quality of English Language

Minor editing of English/spell check is required

Author Response

Largely none responsive to previous comments, the manuscript cannot be recommended for publication in its current state. 1) Improving legibility is not just increasing the dpi but the fonts should be clearly visible and the meaningful presentation; 2) Regarding the target selection, as replied by the authors, the results should be presented (in manuscript and supplementary information) and properly discussed; 3) Showing similar pocket repeatedly in surface rendering is meaningless; 4) Table 2 was not modified as suggested.

Answer: We thank the reviewer for the valuable comments to improve our manuscript, now we have updated the same in the revised manuscript.